# Safety-oriented and explainable machine learning for KSI crash risk prediction: Evidence from the United Kingdom

Khanh Giang Le◉*

Faculty of Civil Engineering, University of Transport and Communications, Hanoi, Vietnam

* gianglk@utc.edu.vn

## Abstract

Road traffic crashes pose a serious public safety challenge, particularly due to fatal and serious injuries. Although machine learning (ML) has been widely used for crash severity prediction, many studies remain accuracy-oriented and insufficiently address class imbalance, decision thresholds, and probabilistic reliability. This study proposes a safety-oriented and explainable ML framework for predicting killed or seriously injured (KSI) crashes using nationwide United Kingdom traffic accident data from 2020–2024. Crash severity is reformulated as a binary classification task distinguishing slight injury crashes from KSI outcomes, aligning model objectives with road safety priorities. A Light Gradient Boosting Machine (LightGBM) model is developed with imbalance handling using SMOTE, safety-oriented decision threshold optimization, and probability calibration. Model performance is evaluated using ROC–AUC, precision–recall analysis, confusion matrices, the Brier score, and a utility-based evaluation metric, while interpretability is ensured through SHapley Additive exPlanations (SHAP). Results show that default threshold settings fail to adequately detect severe crashes. At an optimized threshold of 0.35, the model achieves a Recall(KSI) of 0.605 – representing a substantial 73% improvement compared to conventional configurations – while maintaining acceptable precision. In addition, probability calibration confirms reliable risk estimation (Brier score = 0.190), supporting risk-based interpretation. Comparative analysis demonstrates that the SMOTE-based model provides a more balanced and operationally effective trade-off than class-weighted learning. SHAP analysis identifies speed limit, road class, lighting conditions, and urban context as key variables associated with KSI risk. The findings highlight the importance of safety-oriented learning design and context-aware performance interpretation for effective, risk-based traffic safety management.

**Data availability statement:** All relevant data are publicly available. The UK road traffic crash dataset (STATS19) can be accessed from the UK Department for Transport at: https://www.gov.uk/government/statistical-data-sets/road-safety-open-data. The processed dataset and the code used to generate the results are available in a public repository (Zenodo) at: https://doi.org/10.5281/zenodo.19135778 (dataset) https://doi.org/10.5281/zenodo.19135623 (code).

**Funding:** The author(s) received no specific funding for this work.

**Competing interests:** The authors have declared that no competing interests exist.

## 1. Introduction and related work

Road traffic crashes (RTC) remain a major global public health concern, accounting for approximately 1.19 million fatalities annually and tens of millions of non-fatal injuries [1]. Beyond reducing crash frequency, mitigating crash severity has become a central objective of modern traffic safety research and policy. In this context, accurately identifying high-risk conditions and predicting severe crash outcomes are critical for enabling proactive and evidence-based safety interventions [2].

Traditional regression-based models have been widely used in crash severity analysis; however, they typically rely on predefined functional forms and assumptions of linearity and variable independence. Such assumptions limit their capacity to capture the stochastic and heterogeneous nature of real-world crash processes. As crash datasets have grown in scale and complexity, these limitations have become increasingly pronounced [3–5].

Crash severity is governed by complex and nonlinear interactions among human behavior, roadway characteristics, vehicle dynamics, and environmental conditions. Moreover, the effects of these factors are not static but may vary over time due to behavioral adaptation, environmental variability, and unobserved heterogeneity. Recent studies employing random-parameter logit models have made important advances by capturing temporal instability and seasonal variation in key risk factors, such as lighting conditions, speed limits, and roadway characteristics [6,7]. These findings highlight the inherently dynamic and heterogeneous nature of crash severity, posing challenges for reliable prediction and risk-based safety assessment.

Nevertheless, despite these advances, these econometric approaches are primarily designed for explanatory analysis and inference, and may still face limitations in capturing complex nonlinear interactions and supporting predictive, decision-oriented applications in large-scale and high-dimensional crash datasets. To address these challenges, machine learning (ML) approaches have been increasingly adopted for crash severity prediction due to their ability to model complex nonlinear relationships without strict parametric assumptions. Supervised models, including Decision Trees, Support Vector Machines, Random Forests, and Gradient Boosting methods, have demonstrated strong predictive performance in various traffic safety applications [8–10]. In particular, tree-based ensemble models are well suited to crash data due to their robustness to mixed data types, missing values, and high-dimensional feature spaces. Consequently, ML-based approaches have been widely applied across diverse geographical and roadway contexts [11–13].

Despite these methodological advances, several critical limitations remain. First, RTC datasets are highly imbalanced, with slight injury crashes substantially outnumbering fatal and serious injury cases [14]. As a result, commonly used metrics such as overall accuracy or ROC–AUC may overestimate model performance while masking poor detection of safety-critical KSI crashes [15]. Consequently, models may appear effective while failing to identify a substantial proportion of fatal or serious injury outcomes [8]. Second, many studies adopt default probability thresholds when converting predicted risks into class labels, implicitly assuming equal misclassification costs. This assumption is inappropriate in safety contexts, where failing to identify a

severe crash has far greater consequences than issuing a false alarm [16]. Third, predicted probabilities from ML models are often poorly calibrated, limiting their reliability for risk-based decision-making and policy applications [17]. Finally, although model interpretability has received increasing attention, many high-performing models remain black-box in nature, reducing their practical usefulness for explaining crash mechanisms and supporting targeted interventions [18,19].

The United Kingdom (UK) provides a well-structured and comprehensive data context for addressing these challenges, with standardized crash reporting through the STATS19 system, a hierarchical road classification framework, and heterogeneous traffic regulations [20]. This consistent data infrastructure enables large-scale and reliable analysis of crash severity patterns across regions and time periods. However, studies that simultaneously address class imbalance, decision threshold optimization, probability calibration, and model explainability within the UK context remain limited [21].

To address these gaps, this study proposes a safety-oriented and explainable ML framework for crash severity prediction using nationwide UK crash data from 2020 to 2024. Instead of treating crash severity as a multi-class problem, the analysis is reformulated as a binary classification task distinguishing slight injury crashes from killed or seriously injured (KSI) outcomes. This formulation aligns more closely with practical road safety priorities and reduces ambiguity between fatal and serious injury categories, which are often influenced by post-crash factors beyond roadway conditions. The proposed framework integrates three key components: (i) imbalance-aware learning through data-level resampling techniques, (ii) probability calibration and decision threshold optimization to support risk-based interpretation, and (iii) SHapley Additive exPlanations (SHAP) to enhance model transparency and identify key drivers of severe crash outcomes. The specific objectives of this study are to:

(i) Develop a robust supervised ML model for predicting KSI crash risk using nationwide UK data;

(ii) Evaluate the effectiveness of imbalance-aware learning and threshold optimization in improving the detection of severe crashes;

(iii) Assess the reliability of predicted probabilities through calibration analysis; and

(iv) Provide interpretable insights into the dominant factors influencing severe crash outcomes to support evidence-based traffic safety management.

Unlike much of the existing literature that primarily emphasizes predictive accuracy or algorithmic comparison, this study advances a safety-oriented perspective in crash severity modeling. The main contributions include: (i) a binary reformulation of crash severity focused on KSI outcomes to enhance policy relevance; (ii) a systematic integration of imbalance-aware learning and safety-driven threshold optimization; (iii) the incorporation of probability calibration to improve the reliability of predicted risks; and (iv) the application of SHAP-based explainability to support transparent and policy-relevant insights. Through this integrated framework, crash severity prediction is shifted from accuracy-driven classification toward safety-oriented risk assessment.

## 2. Methodology

### 2.1. Overall framework

This study proposes a safety-oriented supervised machine learning framework for predicting severe traffic crash outcomes on the UK road network. Instead of conventional multi-class severity classification, crash severity is reformulated as a binary classification task distinguishing slight injury crashes from crashes resulting in KSI outcomes. This formulation aligns with practical road safety objectives, where preventing severe human injury is the primary concern.

As illustrated in Fig 1, the proposed framework provides an integrated pipeline that includes data preprocessing, feature engineering, binary target construction, imbalance handling, model training with decision threshold optimization, and probabilistic reliability and explainability analysis. The detailed implementation of this workflow is presented in Algorithm

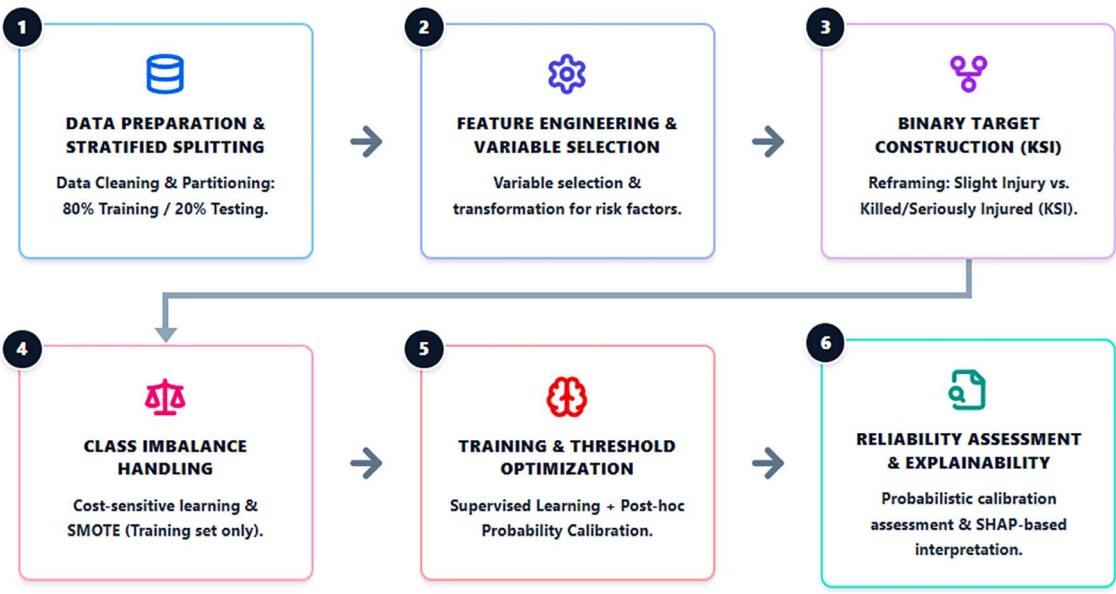

**Fig 1. Safety-oriented supervised machine learning framework.**

1. This unified framework ensures methodological consistency and supports risk-based interpretation for practical traffic safety applications.

### 2.2. Software and implementation environment

All experiments were conducted within a unified computational environment to ensure reproducibility and fair comparison. Table 1 summarizes the software, libraries, and execution platform used in this study.

For transparency, the overall learning procedure is summarized in Algorithm 1.

### Algorithm 1. Safety-oriented KSI crash prediction framework.

**Input:**
Crash dataset $D = \{X, Y\}$, where $Y \in \{0, 1\}$ (0: Slight injury, 1: KSI)
**Output:**
Calibrated KSI probabilities and binary predictions
 1. Remove identifiers and post-crash variables to prevent data leakage.
 2. Encode categorical features, extract temporal components, and impute missing values.
 3. Construct binary target variable (KSI vs. slight injury).
 4. Split $D$ into training (80%) and testing (20%) sets using stratified sampling.
 5. Apply SMOTE to the training set to balance KSI and non-KSI classes.
 6. Train a LightGBM classifier with early stopping on the balanced training data.
 7. Predict KSI probabilities on the test set.
 8. Optimize the decision threshold by maximizing the Operational Utility Score (OUS), which balances Recall (KSI) and Precision under operational constraints.
 9. Evaluate performance using ROC-AUC, PR-AUC, F1-Score, Precision, and Recall metrics.
 10. Assess probability calibration using the Brier Score and calibration curve.
 11. Interpret model predictions using SHAP values.

While Algorithm 1 summarizes the proposed safety-oriented learning pipeline, it is important to clarify how this strategy differs conceptually from default accuracy-oriented learning commonly adopted in crash severity prediction. To this end, Algorithm 2 contrasts the default configuration with the proposed safety-oriented framework.

**Table 1. Summary of software and implementation environment.**

| Component | Description |
|---|---|
| Programming language | Python 3.12 |
| Execution platform | Google Colab |
| Machine learning libraries | LightGBM, XGBoost, scikit-learn |
| Imbalance handling | imbalanced-learn (SMOTE) |
| Model explainability | SHAP |
| Visualization | Matplotlib, Seaborn |
| Hardware environment | Google Colab CPU runtime |

## Algorithm 2. Default accuracy-oriented vs. safety-oriented learning paradigm.

```
Input:
Crash dataset D = {X, Y}, binary target (0: Slight injury, 1: KSI)
Output:
Comparative performance of learning strategies
Default configuration (accuracy-oriented learning)
   1. Split D into training and testing sets using stratified sampling.
   2. Train a LightGBM classifier on the original imbalanced training data.
   3. Optimize model parameters by minimizing the global log-loss.
   4. Apply the default decision threshold τ = 0.5.
   5. Evaluate performance using Accuracy, ROC-AUC, and Recall (KSI).
Safety-oriented configuration (proposed framework)
   6. Apply SMOTE to the training set to balance KSI and non-KSI classes.
   7. Train the proposed classifier (LightGBM) with early stopping on the balanced data.
   8. Predict KSI probabilities on the test set.
   9. Optimize the decision threshold τ by maximizing the OUS, ensuring a balance between detection
sensitivity and false-alarm control.
   10. Evaluate performance using Recall (KSI), Precision-Recall curve, Brier score, and calibra-
tion curve.
```

### 2.3. Data preprocessing and feature engineering

The crash dataset was obtained from the UK Department for Transport and covers five consecutive years (2020–2024), comprising 503,475 crash records with 44 variables [20]. A rigorous preprocessing pipeline was applied to ensure data quality and prevent information leakage (Fig 2).

Feature selection was conducted prior to model training using a domain-informed filtering strategy. This filtering process was designed to eliminate variables that may introduce information leakage and to retain interpretable predictors relevant to traffic safety analysis.

Specifically, variables were systematically evaluated and removed based on three criteria. First, administrative identifiers (e.g., collision_index and collision_ref_no) were excluded because they are non-informative record labels that do not contain meaningful predictive information for crash severity modeling. Second, redundant coordinate systems (e.g., location_easting_osgr and location_northing_osgr) were removed to avoid duplicate spatial representations. Standard geographic coordinates (latitude and longitude) were retained as the primary spatial references, allowing the tree-based model to learn broader spatial patterns and regional safety variations. Finally, post-crash outcome variables (e.g., number_of_casualties and collision_adjusted_severity_serious) were excluded. These post-event variables represent consequences of the crash rather than predictive risk factors, and their inclusion would introduce data leakage into the modeling process.

After this domain-driven filtering, all remaining variables—representing roadway characteristics, environmental conditions, and temporal attributes—were retained without further feature elimination. To further enhance model learning,

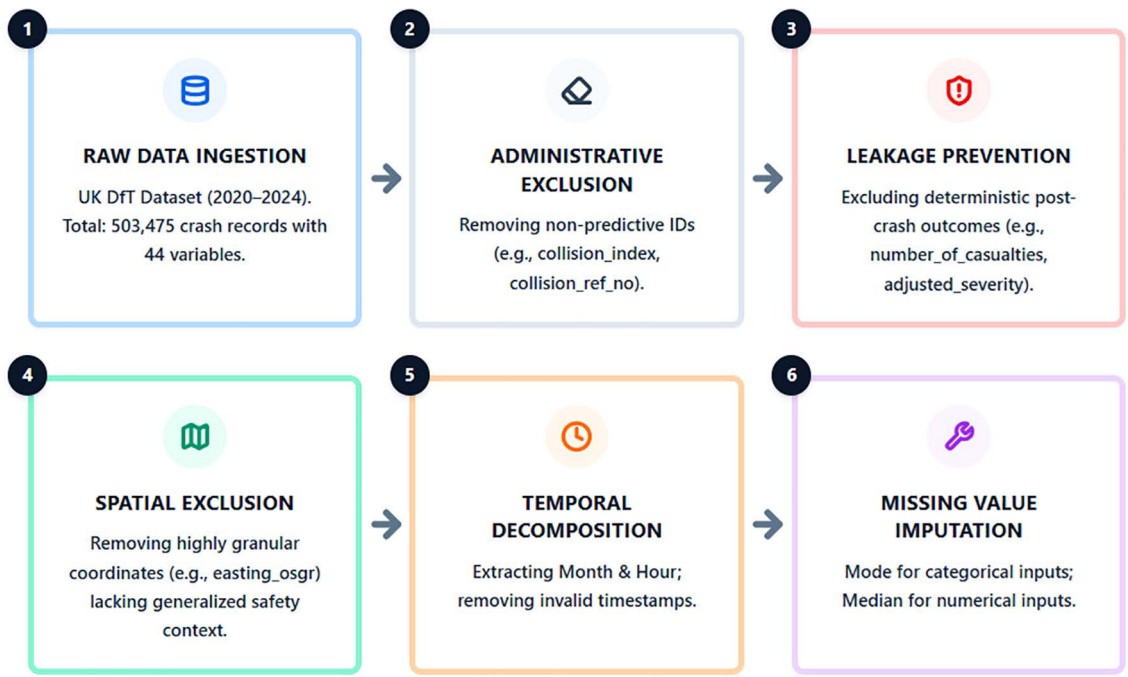

**Fig 2. Data cleaning pipeline and feature extraction strategy.**

temporal variables were decomposed into interpretable components (month and hour of occurrence) to capture seasonal and diurnal traffic patterns. Missing values were subsequently addressed using mode imputation for categorical attributes and median imputation for numerical ones [22]. This preprocessing ensures data consistency and improves model robustness. To strictly prevent information leakage, data splitting was performed prior to any resampling operations [23]. The dataset was divided into training (80%) and testing (20%) subsets using stratified sampling to preserve class proportions [24]. Statistical integrity was verified by comparing class distributions and descriptive statistics between subsets. Ten-fold cross-validation was used within the training set to support model tuning and enhance generalization [25].

## 2.4. Binary target definition: KSI strategy

Crash severity in the STATS19 dataset is recorded as fatal, serious injury, or slight injury [20]. Multi-class prediction suffers from extreme imbalance and ambiguity between fatal and serious outcomes, which are often influenced by stochastic post-crash factors such as emergency response time [26].

Accordingly, this study adopts a binary KSI formulation defined as:

$$y_i = \begin{cases} 1, & \text{if crash } i \in \{Fatal, \ Serious \ Injury\}, \\ 0, & \text{if crash } i = Slight \ Injury. \end{cases} \tag{1}$$

This strategy increases the effective sample size of the minority class and directly reflects the primary safety objective of reducing severe human injury.

## 2.5. Handling class imbalance

Despite the KSI reformulation, class imbalance remains pronounced, with KSI crashes accounting for approximately 23% of observations. To mitigate majority-class bias, data-level imbalance handling was performed using the Synthetic Minority Over-sampling Technique (SMOTE).

SMOTE was applied exclusively to the training set after data splitting, while the test set retained the original class distribution to reflect real-world operating conditions. The resampled training data achieved an approximately balanced 50:50 class distribution, strengthening the learning signal associated with severe crash outcomes.

In addition to SMOTE, class-weighted learning was considered as an alternative imbalance-handling strategy. Class-weighted learning operates at the algorithm level by modifying the loss function to penalize misclassification of the minority class, whereas SMOTE modifies the data distribution through synthetic sample generation.

In this study, SMOTE is adopted as the primary imbalance-handling strategy in the final model. Class-weighted learning is implemented as a comparative approach only, and is not used in the final model configuration. A quantitative comparison between these strategies is presented in the Results section to evaluate their impact on predictive performance and operational effectiveness. The final selection of SMOTE is therefore based on empirical evidence rather than methodological preference.

## 2.6. Model development using LightGBM

Light Gradient Boosting Machine (LightGBM) was selected as the primary predictive model for the proposed safety-oriented framework due to its scalability, robustness to heterogeneous features, and strong performance on large tabular datasets [27,28]. The model was trained using a binary logistic loss function.

Overfitting was controlled through:

- Early stopping based on validation loss (100 rounds);

- Regularization via constrained tree depth and number of leaves; and

- Validation-based monitoring using a hold-out validation subset distinct from the test data.

Categorical variables were handled using LightGBM's native categorical feature support, avoiding extensive one-hot encoding and preserving computational efficiency.

As described in Section 2.5, SMOTE was adopted as the primary imbalance-handling strategy, while class-weighted learning was used only for comparison. Accordingly, the final LightGBM model was trained using SMOTE-balanced data without class weighting. Class-weighted learning is retained as a baseline model, and its performance is analyzed in the Results section. To rigorously evaluate the effectiveness of the proposed LightGBM model, several widely used machine learning algorithms, namely Logistic Regression, Random Forest, and XGBoost, were implemented as baseline models for comparative analysis.

To ensure reproducibility and a fair comparison across all algorithms, the hyperparameters were explicitly configured. For the proposed LightGBM model, the main parameters included a learning rate of 0.03, 1000 boosting iterations, and a tree structure controlled by num_leaves = 40 with an unrestricted maximum depth (max_depth = −1).

For the baseline models, configurations were set as follows: Logistic Regression was implemented with L2 regularization and max_iter = 1000; Random Forest used 100 trees with a maximum depth of 20; and XGBoost was trained with 1000 boosting rounds, learning_rate = 0.03, max_depth = 6, and tree_method = "hist".

These configurations follow commonly adopted settings in tabular machine learning tasks, ensuring both model stability and fair comparison across algorithms.

## 2.7. Decision threshold optimization

Instead of using the default probability threshold (0.50), the decision threshold was optimized to align with safety-oriented objectives, where detecting severe crashes is prioritized over minimizing false alarms. However, threshold selection inherently involves a trade-off between sensitivity (Recall) and predictability (Precision). As emphasized by Xu et al. (2016), lowering the threshold improves the detection of hazardous conditions but inevitably increases the false alarm rate,

reflecting the fundamental trade-off between sensitivity and predictability [29]. From a human factors perspective, excessive false alarms can erode user trust and lead to alarm fatigue, thereby reducing operational effectiveness [30].

To avoid subjective decision-making, threshold selection was formulated as a constrained optimization problem using a penalty-based approach. Predicted KSI probabilities were evaluated across a range of candidate thresholds $\tau \in [0,1]$. An Operational Utility Score (OUS) is defined to maximize Recall while applying a penalty coefficient ($\lambda = 10$) when Precision falls below an operational target ($P_{target} = 0.33$), corresponding to a true-to-false alarm ratio of approximately 1:2. This formulation penalizes excessive false alarms without rigidly discarding near-optimal solutions.

The objective function is defined as follows:

$$OUS(\tau) = Recall(\tau) - \lambda \cdot \max(0, P_{target} - Precision(\tau)) \tag{2}$$

The optimal threshold is therefore selected as:

$$\tau^* = \text{argmax}\_\tau\ OUS(\tau) \tag{3}$$

This formulation explicitly reflects the predictability–sensitivity trade-off described in Xu et al. (2016), where improving sensitivity inevitably increases the false alarm rate [29]. The proposed approach replaces heuristic threshold selection with a data-driven optimization strategy, ensuring a balanced trade-off between detection performance and false-alarm control.

### 2.8. Model evaluation metrics

Model performance was evaluated using ROC–AUC and Precision–Recall AUC (PR–AUC, commonly computed as Average Precision, AP), along with class-specific precision, recall, and F1-score, with particular emphasis on recall for KSI crashes. Precision–Recall curves were used to analyze the trade-off between recall and precision under class imbalance.

### 2.9. Probabilistic reliability and explainability

Probabilistic reliability was assessed using calibration curves and the Brier score, defined as:

$$\text{Brier Score} = \frac{1}{N}\sum_{i=1}^{N}(p_i - y_i)^2 \tag{4}$$

where $p_i$ is the predicted probability of a KSI outcome, $y_i \in \{0, 1\}$ is the observed label, and $N$ is the number of observations. Lower Brier scores indicate better probabilistic reliability, which is essential for risk ranking and intervention prioritization.

Model interpretability was addressed using SHAP. For a given prediction $f(x)$, SHAP expresses the model output as:

$$f(x) = \phi_0 + \sum_{j=1}^{M}\phi_j \tag{5}$$

where $\phi_j$ represents the contribution of feature $j$ to the predicted KSI risk. The magnitude and sign of SHAP values quantify the strength and direction of feature effects, enabling transparent and policy-relevant interpretation.

## 3. Results and discussion

### 3.1. Dataset characteristics and experimental setup

After data cleaning and preprocessing, the final dataset comprised 503,475 crash records described by 30 explanatory variables, reduced from the original 44 variables to prevent data leakage and redundancy. The retained features

represent roadway characteristics, traffic control attributes, environmental conditions, spatial context, and temporal factors.

Following the binary KSI formulation, approximately 76.7% of crashes were classified as slight injury, while 23.3% belonged to the KSI category, confirming a substantial class imbalance typical of national crash datasets. The data were split into training (80%) and testing (20%) subsets using stratified sampling to preserve class proportions. SMOTE was applied exclusively to the training set, while all reported results correspond to the untouched test set.

### 3.2. Default versus safety-oriented learning strategy

To clarify the contribution of the proposed framework, a default learning configuration was first considered. The default configuration follows a conventional accuracy-oriented learning strategy: binary classification without imbalance handling and using the default decision threshold of 0.5. Under this configuration, model optimization is driven by global log-loss, implicitly favoring the majority class (slight injury).

In contrast, the proposed safety-oriented learning strategy integrates data-level imbalance handling (SMOTE) and explicit decision threshold optimization to prioritize the detection of severe crash outcomes. The conceptual differences between these two strategies are summarized in Algorithm 2, and their empirical performance is analyzed in the following subsections.

### 3.3. Overall discriminative performance

**3.3.1. ROC–AUC analysis.** The LightGBM model achieved a ROC–AUC value of 0.664, indicating meaningful discriminative capability. Although moderate in absolute terms, this performance is consistent with values reported in crash severity prediction studies, where a substantial proportion of risk is driven by unobserved human behavior and stochastic factors.

Fig 3 presents the Receiver Operating Characteristic (ROC) curve of the proposed LightGBM model. The curve consistently lies above the diagonal reference line representing random classification, confirming that the model captures statistically significant risk patterns associated with severe crash outcomes. Importantly, ROC–AUC provides a threshold-independent assessment of discriminative performance, establishing a necessary but not sufficient condition for safety-critical deployment.

Previous studies have reported ROC–AUC values in the range of approximately 0.60–0.72 for crash severity prediction under real-world conditions [31,32]. Within this context, the performance achieved in this study falls within the expected range and demonstrates that the proposed model captures statistically meaningful risk patterns despite the inherent uncertainty of crash outcomes.

While ROC–AUC confirms that the model possesses meaningful discriminative capability, it does not prescribe how predicted probabilities should be converted into operational decisions. In safety-critical applications such as KSI detection, the choice of decision threshold plays a decisive role in determining whether severe crashes are effectively identified or overlooked.

**3.3.2. Precision–Recall characteristics under class imbalance.** Given the inherently imbalanced nature of KSI detection, the Precision–Recall (PR) curve provides a more informative performance assessment than ROC analysis. Unlike ROC–AUC, which can remain relatively insensitive to class imbalance, the PR curve directly characterizes the trade-off between detection capability (recall) and false-alarm rate (precision) for rare but safety-critical events.

Fig 4 illustrates the Precision–Recall curve, demonstrating that meaningful recall levels can be maintained across a wide range of precision values, with a PR–AUC of 0.35. This result confirms that the model retains substantial discriminative power for detecting KSI crashes despite the skewed class distribution.

Importantly, the PR curve reveals that increasing recall beyond conventional thresholds is feasible, albeit at the cost of reduced precision. This observation directly motivates the adoption of a safety-oriented decision threshold, where sensitivity to severe crashes is prioritized over overall accuracy.

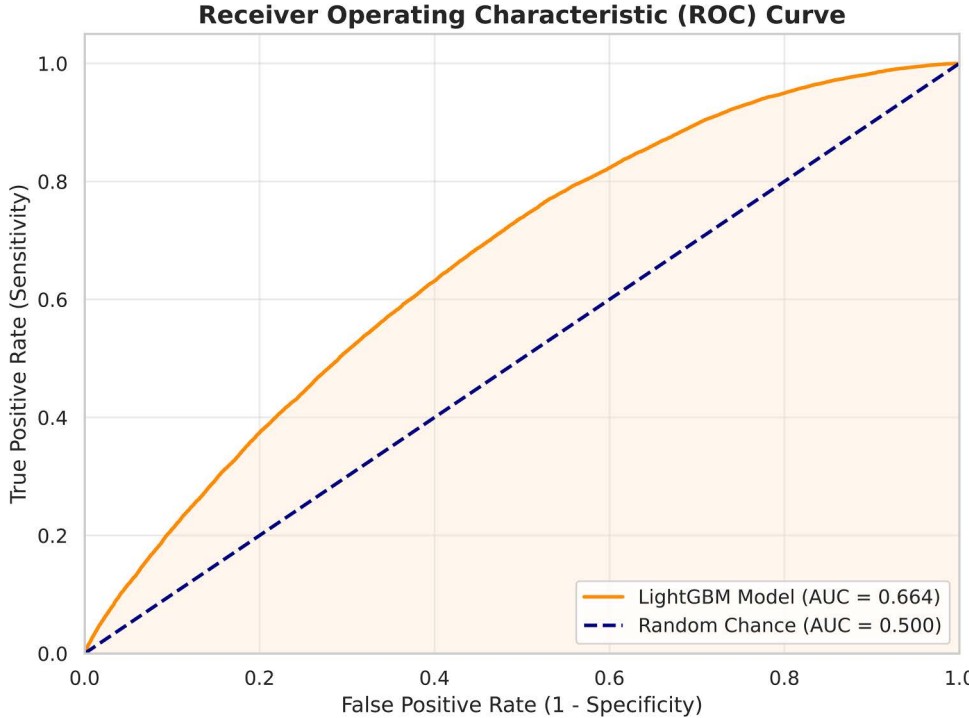

**Fig 3. ROC curve illustrating the discriminative ability of the LightGBM model for KSI crash risk prediction.**

The PR analysis highlights that the default probability threshold of 0.5 is not necessarily optimal for KSI detection. Instead, it reveals a continuum of operating points where recall can be substantially improved with an acceptable increase in false positives. This insight motivates the explicit optimization of the decision threshold from a safety-first perspective.

**3.3.3. Decision threshold optimization and confusion matrix analysis.** Building on the threshold-independent insights provided by the ROC and Precision–Recall analyses, the decision threshold was explicitly optimized to align model behavior with safety-oriented objectives. In crash severity prediction, the cost of missing a severe crash (false negative) is substantially higher than that of issuing a false alarm (false positive). Consequently, the default probability threshold of 0.5—typically optimized for overall accuracy—was deemed inappropriate for KSI detection.

A range of probability thresholds was systematically evaluated on the validation set. The optimal operating point was identified at $\tau^* = 0.35$, which provides a favorable balance between sensitivity to KSI crashes and practical precision. To provide a quantitative justification for the threshold selection, a sensitivity analysis was conducted across candidate thresholds ranging from 0.20 to 0.50. The results are summarized in Table 2.

As shown in Table 2, lower thresholds ($\tau \leq 0.30$) achieve high recall (0.699–0.867) but incur heavy penalties due to low precision, resulting in reduced OUS values and indicating excessive false alarm rates. In contrast, higher thresholds ($\tau \geq 0.40$) avoid penalties but exhibit a significant reduction in recall (down to 0.349–0.514), limiting the model's ability to detect severe crashes.

The threshold $\tau^* = 0.35$ emerges as the optimal operating point. Its precision (0.328) is slightly below the target (0.33), incurring only a minimal penalty. In exchange, the model retains a relatively high recall of 0.605, leading to the maximum OUS (0.586). This result indicates that $\tau^* = 0.35$ provides an effective data-driven trade-off between detection capability and false-alarm control for practical deployment.

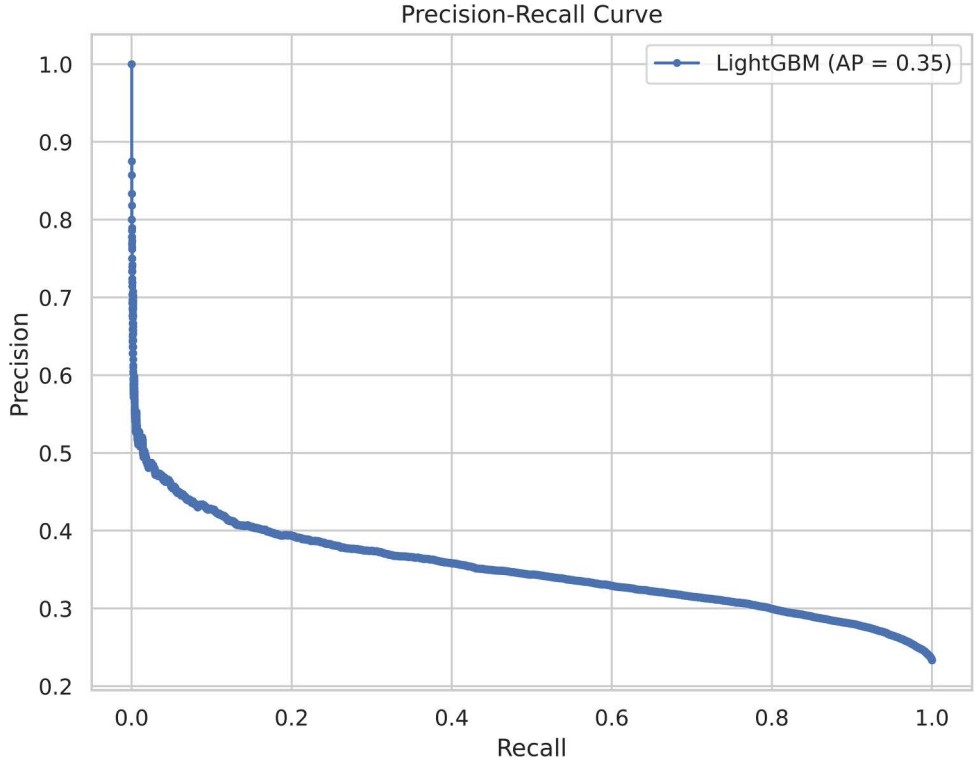

**Fig 4. Precision–Recall curve of the LightGBM model for KSI crash detection under class imbalance.**

**Table 2. Quantitative threshold optimization based on sensitivity–predictability trade-off.**

| Threshold (τ) | Precision | Recall | OUS |
|---|---|---|---|
| 0.20 | 0.286 | 0.867 | 0.430 |
| 0.25 | 0.302 | 0.790 | 0.511 |
| 0.30 | 0.315 | 0.699 | 0.554 |
| **0.35** | **0.328** | **0.605** | **0.586** |
| 0.40 | 0.342 | 0.514 | 0.514 |
| 0.50 | 0.366 | 0.349 | 0.349 |

At the selected optimal threshold (τ* = 0.35), the LightGBM model achieves a Recall for KSI crashes of 0.605. This represents a substantial 73% improvement compared to the default threshold configuration (τ = 0.50), where the recall value is only 0.349. From a road safety perspective, this result implies that the proposed framework is capable of identifying approximately 60% of severe crashes, significantly enhancing the potential for proactive safety interventions.

The corresponding Precision (KSI) of 0.328 reflects an intentional and acceptable trade-off in a safety-first context. Many false-positive predictions may correspond to high-risk conditions that did not result in severe injury due to stochastic or contextual factors. Flagging such cases remains valuable for preventive planning, infrastructure improvement, and risk-based prioritization.

Table 3 summarizes the classification performance of the LightGBM model at the optimized decision threshold ($\tau^* = 0.35$), highlighting the trade-off between sensitivity and precision under the proposed safety-oriented decision strategy.

The practical implications of this trade-off are further illustrated by the confusion matrix in Fig 5. Compared with the default threshold, the optimized configuration substantially reduces false negatives for the KSI class, confirming that threshold tuning is a critical mechanism for aligning machine learning outputs with real-world road safety priorities rather than purely statistical optimality.

**3.3.4. Comparative performance with baseline models.** Before comparing different machine learning algorithms, it is important to clarify the impact of imbalance-handling strategies on model performance. To improve transparency, a comparative analysis between SMOTE and class-weighted learning was conducted under the same decision threshold ($\tau^* = 0.35$). The results are presented in Table 4.

**Table 3. Classification performance of the LightGBM model at the optimized threshold ($\tau^* = 0.35$).**

| Metric | Slight (Non-severe) | KSI (Fatal/Serious) |
|---|---|---|
| Precision | 0.838 | 0.328 |
| Recall | 0.623 | 0.605 |
| F1-score | 0.715 | 0.425 |

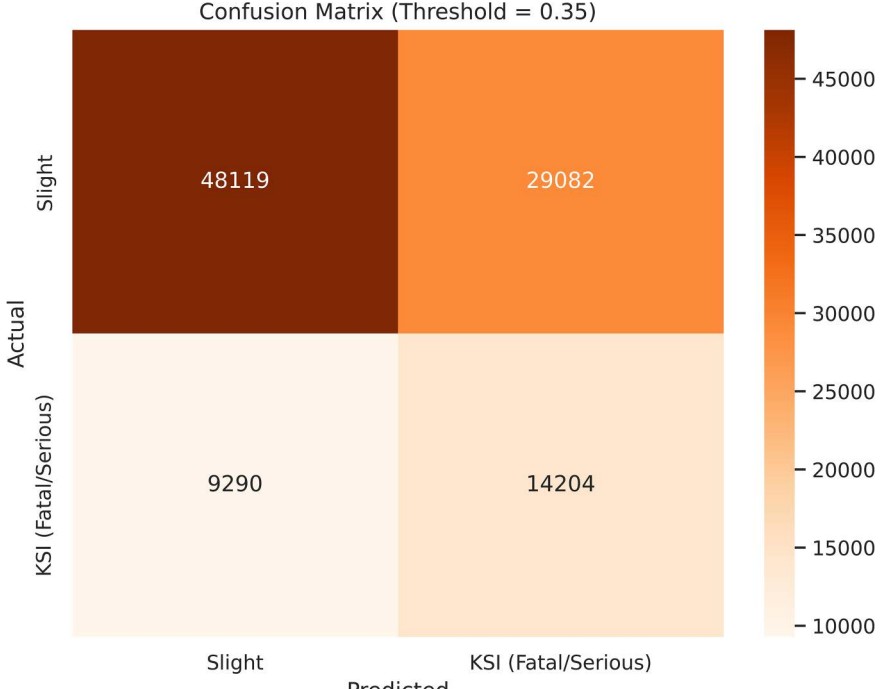

**Fig 5. Confusion matrix of the LightGBM model at the optimized decision threshold ($\tau = 0.35$), illustrating the trade-off between increased KSI detection and higher false-positive rates under a safety-oriented learning strategy.**

**Table 4. Comparison of imbalance-handling strategies.**

| Imbalance Strategy | ROC–AUC | Brier Score ↓ | Precision (KSI) | Recall (KSI) | OUS |
|---|---|---|---|---|---|
| Class-Weighted Learning | 0.692 | 0.221 | 0.284 | 0.904 | 0.445 |
| SMOTE (Proposed) | 0.664 | 0.190 | 0.328 | 0.605 | 0.586 |

As shown, class-weighted learning achieves a substantially higher Recall (0.904), indicating strong sensitivity to KSI crashes. However, this comes at the expense of a significantly reduced Precision (0.284), reflecting an excessive false alarm rate. In safety-critical applications, such behavior may lead to alarm fatigue and reduced operational trust.

In contrast, the SMOTE-based model provides a more balanced performance, achieving a higher Precision (0.328) while maintaining a reasonable Recall (0.605). In addition, SMOTE yields a substantially lower Brier score (0.190 vs. 0.221), indicating more reliable probability estimates for risk-based decision-making. Importantly, when evaluated using the proposed Operational Utility Score (OUS), which explicitly incorporates both recall and precision constraints, SMOTE achieves the highest utility (0.586), outperforming class-weighted learning (0.445).

These results indicate that although class-weighted learning maximizes sensitivity, it does so at the cost of operational feasibility. SMOTE, on the other hand, provides a more effective trade-off between detection capability and false-alarm control, making it more suitable for real-world deployment in traffic safety systems.

Table 5 presents the comparative performance of the evaluated machine learning models at the safety-optimized decision threshold ($\tau^* = 0.35$). The comparison includes the proposed LightGBM model and three widely used baseline algorithms: Logistic Regression, Random Forest, and XGBoost.

As commonly observed in crash severity modeling, no single algorithm dominates all evaluation metrics due to the inherent trade-off between precision and recall. For example, Logistic Regression achieved the highest Recall (0.808), but this occurred at the expense of substantially lower Precision (0.282), implying a high rate of false alarms.

Among the evaluated models, LightGBM achieved the highest ROC–AUC (0.664), indicating the highest overall discriminative performance among the evaluated models. In addition, LightGBM achieved the highest Precision for KSI detection (0.328) while maintaining competitive Recall and F1-score values. The PR–AUC values are also comparable across the evaluated models, with XGBoost and LightGBM achieving the highest scores (approximately 0.35), indicating robust performance under class imbalance.

Importantly, LightGBM also produced the lowest Brier score (0.190), indicating superior probabilistic reliability compared with Random Forest (0.201) and XGBoost (0.192). These results suggest that the LightGBM model provides a balanced combination of discrimination and probabilistic reliability, making it particularly suitable for safety-oriented crash risk prediction.

The comparative analysis confirms that the proposed LightGBM model provides a strong balance between discriminative performance and probabilistic reliability at the safety-optimized threshold. Building on this foundation, the next section further investigates the model's probabilistic reliability through calibration analysis, which is essential for risk-based traffic safety management.

**Table 5. Comparative performance of machine learning models for KSI crash prediction at the safety-optimized decision threshold ($\tau^* = 0.35$).**

| Model | ROC–AUC | PR–AUC | Brier Score ↓ | Precision (KSI) | Recall (KSI) | F1-score |
|---|---|---|---|---|---|---|
| Logistic Regression | 0.636 | 0.334 | 0.227 | 0.282 | 0.808 | 0.418 |
| Random Forest | 0.660 | 0.345 | 0.201 | 0.311 | 0.704 | 0.432 |
| XGBoost | 0.663 | 0.350 | 0.192 | 0.325 | 0.621 | 0.427 |
| LightGBM (Proposed) | 0.664 | 0.349 | 0.190 | 0.328 | 0.605 | 0.425 |

### 3.3.5. Contextual interpretation of model performance.

Crash severity prediction is inherently challenging due to the complex and stochastic nature of road traffic crashes. Unlike many prediction tasks in controlled domains, traffic accidents are influenced by numerous interacting factors, including roadway characteristics, driver behavior, vehicle conditions, environmental factors, and random circumstances occurring at the moment of collision. As highlighted by Savolainen et al. (2011), crash severity outcomes are strongly affected by unobserved heterogeneity and behavioral uncertainty, which significantly limits the achievable predictive accuracy of statistical and machine learning models [26]. Consequently, moderate predictive performance is commonly reported in the crash severity literature.

Empirical evidence from recent machine learning studies supports this observation. For instance, Komol et al. (2021) applied Random Forest, Support Vector Machine, and K-Nearest Neighbor models to analyze crash severity of vulnerable road users using Queensland crash data and reported test accuracies ranging from 64.45% to 72.30%, depending on the road user group considered [31]. Similarly, Rifat et al. (2024) reported that a recent LightGBM-based fatality prediction model achieved a maximum ROC–AUC of approximately 0.72 when comparing multiple machine learning algorithms for traffic accident fatality classification [32]. These findings suggest that predictive performance in the range of AUC ≈ 0.60– 0.72 is commonly observed in real-world crash severity modeling problems.

Within this context, the ROC–AUC value obtained in this study (0.664) is consistent with the performance range reported in the literature [31,32]. Rather than being interpreted against a universal benchmark, this result should be understood in relation to the inherent complexity and stochastic nature of crash severity prediction.

Accordingly, the focus of this study is not solely on maximizing global discrimination metrics, but on developing a safety-oriented modeling framework. By optimizing the decision threshold to 0.35, the proposed LightGBM model achieves a Recall of 0.605 for the KSI class, successfully identifying a substantial proportion of severe crashes despite class imbalance.

This perspective highlights that model performance in crash severity prediction should be interpreted in a context-aware manner, where moderate ROC–AUC values can still provide meaningful and actionable insights when aligned with safety-critical objectives and decision-making requirements.

## 3.4. Probability calibration and reliability analysis

Beyond classification accuracy, reliable probability estimates are essential for risk-based traffic safety management, where predicted risks are used to prioritize locations, conditions, or time periods for intervention rather than merely assigning class labels.

The probabilistic performance of the proposed LightGBM model was evaluated using the Brier score and calibration (reliability) curve. The Brier score measures the mean squared error between predicted probabilities and observed outcomes, and lower scores indicate better probabilistic reliability [33,34]. The model achieved a Brier score of 0.190, indicating a relatively low mean squared error between predicted KSI probabilities and observed outcomes. For rare-event prediction problems such as fatal and serious injury crashes, a Brier score below 0.20 is generally considered indicative of good probabilistic reliability.

Fig 6 presents the calibration curve of the LightGBM model. Each point represents the empirical frequency of KSI crashes within a predicted probability bin, plotted against the mean predicted probability. The dashed diagonal line denotes perfect calibration. Overall, the predicted probabilities closely follow the ideal diagonal across most probability ranges, indicating that the model's outputs are well calibrated. Minor deviations are observed at higher probability levels, where the model exhibits slight under-confidence. Such behavior is expected in crash severity prediction due to the inherent stochasticity of severe injury outcomes and the limited number of extreme-risk observations.

Importantly, this level of calibration implies that the model's probabilistic outputs are interpretable and trustworthy. For example, a predicted KSI probability of 0.6 can be meaningfully interpreted as an approximately 60% empirical risk, rather than an arbitrary confidence score. This property is critical for risk ranking, early warning systems, and evidence-based

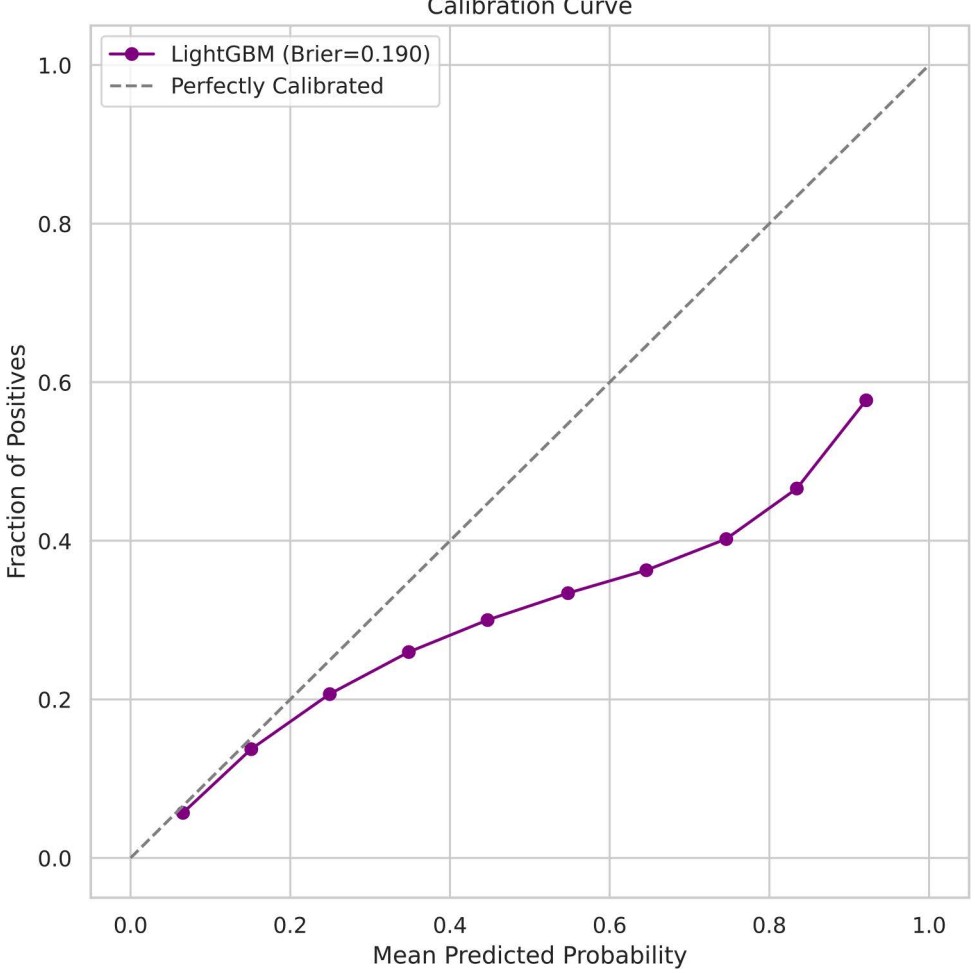

**Fig 6. Calibration curve of the LightGBM model for KSI prediction.** The close alignment between predicted probabilities and observed frequencies indicates good probabilistic reliability, supporting risk-based interpretation and prioritization of traffic safety interventions.

allocation of traffic safety resources. In contrast to uncalibrated boosting models, which often produce overconfident probabilities, the calibrated LightGBM framework supports policy-oriented decision-making, enabling authorities to move beyond binary predictions toward probabilistic, risk-informed safety management.

While the calibration analysis confirms that the proposed model provides reliable and interpretable probability estimates for risk-based decision-making, understanding why specific crashes are assigned higher KSI risk remains essential. Therefore, the following section employs SHAP-based explainability to uncover the key factors driving severe crash predictions and to translate model outputs into actionable safety insights.

### 3.5. Model explainability using SHAP

In the SHAP summary plot (Fig 7), input features are ranked in descending order according to their mean absolute SHAP values across all observations. This ranking reflects the global contribution of each variable to the model's predictions, irrespective of whether its effect increases or decreases KSI risk. Variables appearing at the top of the plot therefore exert the strongest overall influence on crash severity prediction within the learned model.

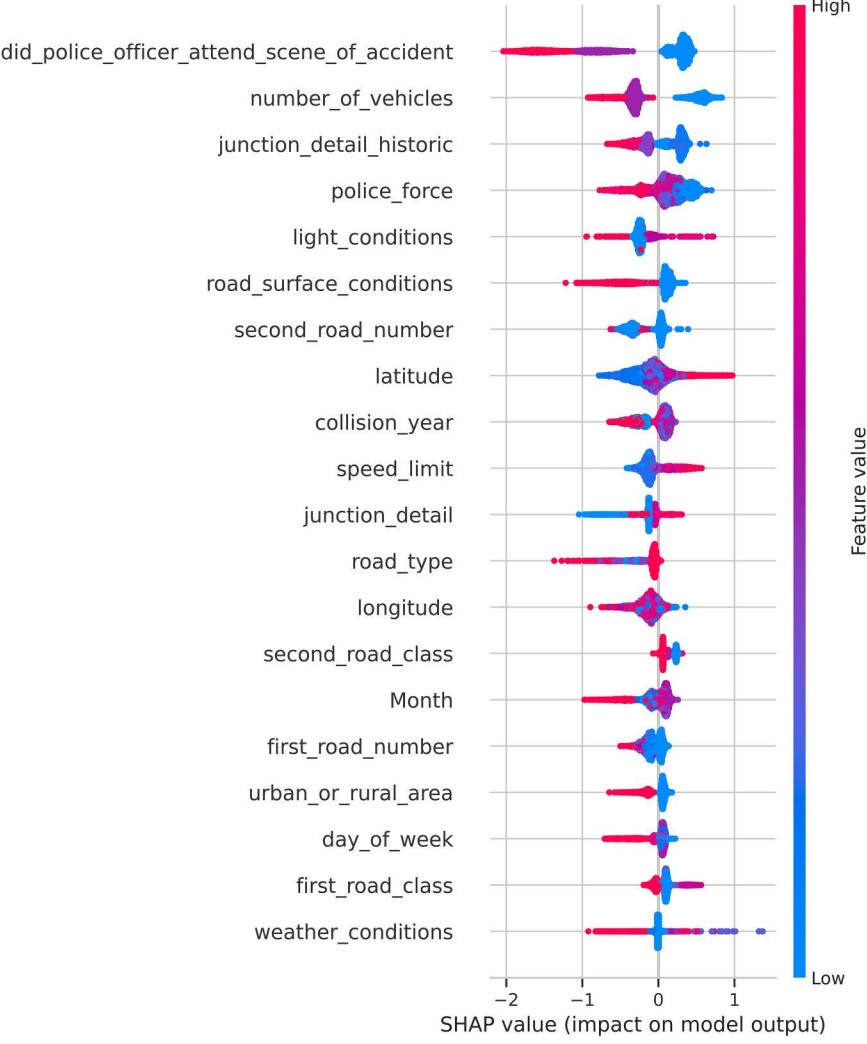

**Fig 7. SHAP summary plot illustrating the encoded-feature contributions to KSI crash risk prediction.** Note: Feature colors indicate higher (red) and lower (blue) encoded values according to the STATS19 coding scheme. SHAP values represent conditional contributions relative to the model baseline and do not imply causality.

It should be emphasized that this ordering represents model-based importance rather than causal dominance. SHAP values quantify conditional contributions learned from the data and are inherently dependent on the observed feature distributions and the underlying model structure.

Fig 7 visualizes the direction and magnitude of feature contributions to KSI risk across the entire dataset. To facilitate interpretation and translate these patterns into domain-relevant insights, Table 6 summarizes the SHAP patterns of the most influential explanatory variables, together with their corresponding STATS19 coding schemes and safety interpretations.

It is important to explicitly distinguish between statistical associations identified by the model and actionable safety mechanisms. SHAP values capture patterns present in the observed data, which may be influenced by reporting practices, exposure differences, or behavioral adaptations, rather than direct causal effects.

**Table 6. Interpretation of SHAP patterns for selected explanatory variables (non-causal, risk-oriented interpretation).**

| Variable | STATS19 coding | SHAP pattern | Interpretation |
|---|---|---|---|
| did_police_officer_ attend_scene_of _accident | No = 1; Yes = 2 | Police attendance→SHAP<0 | Crashes without recorded police attendance are more frequently associated with higher predicted KSI risk, which may reflect reporting or post-event factors rather than causal effects. |
| number_of_vehicles | 1, 2, 3, … | Higher values→SHAP<0 | Single-vehicle crashes are more frequently associated with higher predicted KSI risk compared with multi-vehicle crashes. |
| road_type | Roundabout = 1→One way street→Dual carriageway→Single carriageway→Slip road→One way street/Slip road = 7 | Certain types (Slip road, One-way street/Slip road) → SHAP<0 | Certain road types are associated with lower predicted KSI risk, reflecting differences in geometric design and traffic conditions. |
| road_surface_conditions | Dry = 1→Wet = 2; Snow = 3; Ice→Flood→Oil→Mud = 7 | Better conditions→SHAP>0 | Better road surface conditions are associated with higher predicted KSI risk, which may reflect higher travel speeds under favorable conditions. |
| speed_limit | 20→70 mph | Higher values→ SHAP>0 | Higher speed limits are strongly associated with increased predicted KSI risk. |
| junction_detail_historic | Roundabout = 1; T junction→ Slip road→Crossroads→>4 arms = 7 | More complex junctions→SHAP<0 | More complex junction layouts are associated with lower predicted KSI risk, potentially reflecting reduced speeds and increased driver caution. |
| first_road_class/ second_road_class | 1: Motorway→2: A(M) → 3: A→4: B→5: C→6: Unclassified | Lower-class roads→SHAP>0 | Lower-class roads are associated with higher predicted KSI risk, possibly due to roadway design and protection differences. |
| light_conditions | Daylight = 1→Darkness = 7 | Darkness→SHAP>0 | Darkness is associated with higher predicted KSI risk, reflecting reduced visibility conditions. |
| weather_conditions | Fine = 1→Rain→Snow→Snowing + high winds →Fog = 7 | Higher values→SHAP<0 | Adverse weather conditions are associated with lower predicted KSI risk, which may reflect behavioral adaptation such as reduced speed. |
| urban_or_rural_area | Urban = 1; Rural = 2 | Urban→SHAP>0 | Urban areas are associated with higher predicted KSI risk in the observed data. |

For example, variables such as police attendance or weather conditions may reflect post-event reporting processes or behavioral responses (e.g., reduced speed under adverse conditions), rather than intrinsic risk factors. Therefore, the observed SHAP patterns should be interpreted as associative signals that help identify high-risk conditions, rather than as direct evidence of causal relationships.

From a practical perspective, these insights are most appropriately used for risk screening and prioritization, while causal interpretation and policy intervention should be supported by domain knowledge and complementary analytical approaches.

Despite its advantages in enhancing model transparency, SHAP-based interpretation has several inherent limitations that should be acknowledged. First, SHAP values describe conditional associations learned from observational data rather than causal relationships, and therefore should not be interpreted as evidence of direct causality. Second, some observed patterns may be influenced by reporting biases or behavioral adaptations rather than underlying risk mechanisms. For example, variables related to police attendance or weather conditions may reflect differences in reporting practices or driver responses rather than intrinsic crash severity determinants. Accordingly, the SHAP results in this study are intended to support risk interpretation and prioritization, while causal inference and policy design should rely on complementary domain-specific analyses.

## 3.6. Integrated discussion and practical implications

This study demonstrates that effective traffic crash severity prediction depends less on algorithmic complexity than on the strategic alignment of the learning process with safety-critical objectives. Conventional accuracy-oriented machine learning models, even when achieving acceptable ROC–AUC values, tend to underperform in detecting fatal and serious injury crashes due to class imbalance, inappropriate decision thresholds, and unreliable probability estimates. The proposed safety-oriented framework addresses these limitations by integrating imbalance-aware learning, decision threshold optimization, probabilistic calibration, and explainable modeling within a unified pipeline.

From a methodological perspective, the results confirm that threshold optimization plays a decisive role in safety-critical classification. By shifting the decision threshold from the conventional value of 0.5 to an optimized value of 0.35, the model doubled the recall of KSI crashes without relying on more complex algorithms or additional data sources. This finding highlights that many previously reported limitations in crash severity prediction may stem from decision strategy design rather than intrinsic model capability.

The comparison of imbalance-handling strategies further highlights that maximizing recall alone is insufficient for safety-critical applications. A balanced trade-off between detection performance and false-alarm control is essential, reinforcing the importance of integrating data-level strategies such as SMOTE with decision-aware evaluation metrics.The SHAP-based analysis provides further insight into the mechanisms underlying severe crash risk. Dominant contributors such as speed limit, road classification, lighting conditions, and road geometry are consistent with established traffic safety theory. Meanwhile, several counterintuitive patterns—such as higher KSI risk under good weather or dry road conditions—warrant further scrutiny. These patterns have been widely discussed in the traffic safety literature and are commonly interpreted through behavioral adaptation and risk compensation mechanisms [35,36]. According to risk compensation theory, drivers may adjust their behavior to maintain an acceptable level of perceived risk. For instance, under adverse weather or slippery road conditions, drivers typically perceive higher risk and may compensate by driving more cautiously, often reducing travel speeds. As a result, although crash frequency may increase under such unfavorable conditions, previous studies have reported that crash severity may be mitigated by lower impact speeds [37]. Conversely, favorable environmental conditions can create a false sense of safety, encouraging higher travel speeds and reduced vigilance. When collisions occur under these conditions, the higher kinetic energy associated with greater travel speeds substantially increases the likelihood of severe outcomes, including KSI crashes. Importantly, these patterns reflect conditional risk relationships learned from observational data rather than direct causal effects, highlighting the importance of careful interpretation when translating model outputs into safety policies.

From a practical standpoint, the calibrated probability outputs enable the proposed model to function as a risk-ranking and early-warning tool, rather than a simple classifier. Transportation authorities can use predicted KSI probabilities to prioritize safety interventions across road types, time periods, or environmental conditions, supporting proactive and evidence-based decision-making. The explicit trade-off between recall and precision is particularly relevant for safety management, where missing severe crashes is typically more costly than issuing precautionary warnings for high-risk but non-severe events.

To operationalize these safety-oriented thresholds, transportation authorities could integrate the proposed framework into real-time traffic management systems. In practice, contextual crash information (e.g., location, roadway characteristics, weather conditions, and traffic environment) can be rapidly processed by the model to estimate the probability of a severe crash outcome. When the predicted probability exceeds a predefined high-risk threshold, the system may automatically trigger an alert within Traffic Management Centers (TMCs) or emergency coordination platforms.

Such threshold-based alerts can support proactive decision-making, including issuing driver warnings through variable message signs, prioritizing incident management resources, or coordinating emergency response services. By translating model predictions into operational alert levels, the proposed framework provides a practical mechanism for transforming predictive analytics into actionable traffic safety interventions.

Finally, it is important to acknowledge that the study period (2020–2024) partially overlaps with the COVID-19 pandemic, during which traffic demand and travel behavior experienced temporary disruptions in many regions. Reduced traffic volumes, changes in travel purposes, and altered mobility patterns during certain pandemic phases may have influenced the distribution of crash characteristics used for model training. However, the dataset spans multiple years and includes both pandemic and post-pandemic traffic conditions, which helps mitigate the influence of short-term anomalies. Consequently, the model is more likely to capture broader crash severity patterns rather than behaviors specific to pandemic-related travel conditions. Future research could further examine the potential impact of pandemic-induced mobility changes through more detailed temporal segmentation of crash data.

## 4. Conclusion

This study proposed a safety-oriented and explainable machine learning framework for predicting killed or seriously injured (KSI) traffic crashes using five years of nationwide UK crash data. By reformulating crash severity prediction as a binary KSI detection problem, the framework aligns model objectives with real-world traffic safety priorities and mitigates ambiguity associated with multi-class severity labels.

The results demonstrate that strategic learning design plays a more critical role than algorithmic complexity. Through imbalance-aware training using SMOTE and explicit decision threshold optimization, the proposed framework achieved a Recall(KSI) of 0.605—representing a substantial improvement over conventional accuracy-oriented configurations—while maintaining an operationally acceptable precision level.

The achieved ROC–AUC of 0.664 indicates meaningful discriminative capability and is consistent with performance ranges reported in prior crash severity prediction studies, where predictive performance is inherently constrained by behavioral uncertainty and stochastic factors. At the same time, the calibrated probability estimates (Brier score = 0.190) ensure that predicted risks are reliable and interpretable for risk-based decision-making.

A comparative analysis of imbalance-handling strategies further shows that, although class-weighted learning maximizes sensitivity, it leads to excessive false alarms and reduced operational feasibility. In contrast, the SMOTE-based model provides a more balanced trade-off between detection capability and false-alarm control, achieving the highest operational utility under the proposed evaluation framework.

SHAP-based explainability enhances model transparency by identifying key variables associated with KSI risk, including speed limit, road classification, lighting conditions, and environmental context. These findings should be interpreted as statistical associations rather than causal relationships, and are most appropriately used to support risk screening and prioritization in traffic safety management.

Despite these contributions, several limitations should be acknowledged. The analysis relies on historical crash data and does not incorporate real-time traffic flow, vehicle dynamics, or behavioral sensing data. In addition, spatial dependence and network-level interactions were not explicitly modeled. Future research may integrate spatiotemporal learning approaches and real-time data sources to further improve predictive performance and operational deployment.

In conclusion, this research advances machine learning applications in traffic safety by shifting the focus from accuracy-driven classification toward safety-oriented, interpretable, and probabilistically reliable risk prediction, providing a practical foundation for proactive and evidence-based traffic safety management.

## Supporting information

**S1 File. Code for safety-oriented and explainable machine learning for KSI crash risk prediction.**
(RAR)

**S2 File. Road traffic crash data in the United Kingdom (2020–2024).**
(RAR)

## Author contributions

**Conceptualization:** Khanh Giang Le.

**Data curation:** Khanh Giang Le.

**Formal analysis:** Khanh Giang Le.

**Investigation:** Khanh Giang Le.

**Methodology:** Khanh Giang Le.

**Software:** Khanh Giang Le.

**Supervision:** Khanh Giang Le.

**Validation:** Khanh Giang Le.

**Visualization:** Khanh Giang Le.

**Writing – original draft:** Khanh Giang Le.

**Writing – review & editing:** Khanh Giang Le.

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
