## [Decision Letter · Decision Letter 0]

4 Feb 2026

PONE-D-26-00177Safety-Oriented and Explainable Machine Learning for KSI Crash Risk Prediction: Evidence from the United KingdomPLOS One

Dear Dr. Le,

Thank you for submitting your manuscript to PLOS ONE. After careful consideration, we feel that it has merit but does not fully meet PLOS ONE’s publication criteria as it currently stands. Therefore, we invite you to submit a revised version of the manuscript that addresses the points raised during the review process.

We look forward to receiving your revised manuscript.

Kind regards,

Xuecai Xu, Ph.D.

Academic Editor

PLOS One

Journal Requirements:

3. Thank you for uploading your study's underlying data set. Unfortunately, the repository you have noted in your Data Availability statement does not qualify as an acceptable data repository according to PLOS's standards.

4. We note you have included a table to which you do not refer in the text of your manuscript. Please ensure that you refer to Table 3 in your text; if accepted, production will need this reference to link the reader to the Table.

Additional Editor Comments:

Please make the revision carefully as the reviewers suggest.

Reviewers' comments:

Reviewer's Responses to Questions

**Comments to the Author**

1. Is the manuscript technically sound, and do the data support the conclusions?

Reviewer #1: Yes

Reviewer #2: Yes

2. Has the statistical analysis been performed appropriately and rigorously? 

Reviewer #1: Yes

Reviewer #2: Yes

3. Have the authors made all data underlying the findings in their manuscript fully available?

The PLOS Data policy requires authors to make all data underlying the findings described in their manuscript fully available without restriction, with rare exception (please refer to the Data Availability Statement in the manuscript PDF file). The data should be provided as part of the manuscript or its supporting information, or deposited to a public repository. For example, in addition to summary statistics, the data points behind means, medians and variance measures should be available. If there are restrictions on publicly sharing data—e.g. participant privacy or use of data from a third party—those must be specified.requires authors to make all data underlying the findings described in their manuscript fully available without restriction, with rare exception (please refer to the Data Availability Statement in the manuscript PDF file). The data should be provided as part of the manuscript or its supporting information, or deposited to a public repository. For example, in addition to summary statistics, the data points behind means, medians and variance measures should be available. If there are restrictions on publicly sharing data—e.g. participant privacy or use of data from a third party—those must be specified.requires authors to make all data underlying the findings described in their manuscript fully available without restriction, with rare exception (please refer to the Data Availability Statement in the manuscript PDF file). The data should be provided as part of the manuscript or its supporting information, or deposited to a public repository. For example, in addition to summary statistics, the data points behind means, medians and variance measures should be available. If there are restrictions on publicly sharing data—e.g. participant privacy or use of data from a third party—those must be specified.requires authors to make all data underlying the findings described in their manuscript fully available without restriction, with rare exception (please refer to the Data Availability Statement in the manuscript PDF file). The data should be provided as part of the manuscript or its supporting information, or deposited to a public repository. For example, in addition to summary statistics, the data points behind means, medians and variance measures should be available. If there are restrictions on publicly sharing data—e.g. participant privacy or use of data from a third party—those must be specified.

Reviewer #1: Yes

Reviewer #2: Yes

4. Is the manuscript presented in an intelligible fashion and written in standard English?

Reviewer #1: Yes

Reviewer #2: Yes

5. Review Comments to the Author

Reviewer #1: Overall Recommendation: Minor Revision

The manuscript proposes a safety-oriented and explainable machine learning (ML) framework for predicting Killed or Seriously Injured (KSI) crashes using nationwide UK traffic accident data from 2020 to 2024. By framing the problem as a binary classification (KSI vs. slight injury) and systematically addressing class imbalance, decision threshold optimisation, probability calibration, and model interpretability, the study offers a meaningful contribution to the field of traffic safety analytics. However, several issues should be addressed to strengthen before final acceptance.

1. Please clarify whether feature selection was performed before modelling.

2. Please clarify the hyperparameter settings for the adopted models.

3. The reported ROC-AUC value is 0.664. The authors could provide a more apparent justification for why this level is sufficient for safety-critical applications, perhaps by citing similar studies in the literature.

4. There is no direct comparison with baseline models (e.g., logistic regression, random forest, XGBoost).

5. The finding that “good road surface conditions” and “fine weather” are associated with higher KSI risk is intriguing but requires deeper scrutiny. The authors could find support from previous studies.

6. The authors could emphasise the linkage between safety-oriented thresholds and real-world intervention strategies. Please guide the transport authorities on operationalising threshold-based alerts.

7. The data covers 2020–2024, which includes the COVID-19 pandemic period. The authors should briefly discuss whether and how pandemic-induced changes in traffic patterns might affect model training.

Reviewer #2: Literature review could be strengthened.

The related work section would benefit from a clearer synthesis of prior studies on imbalance handling, threshold selection, and probability calibration in crash severity modeling such as https://doi.org/10.1016/j.ijtst.2024.11.009 and https://doi.org/10.1016/j.aap.2025.108277. Explicitly positioning this study relative to those strands would better highlight its contribution.

Decision threshold optimization needs clearer justification.

The choice of optimizing Recall(KSI) is reasonable, but the practical constraint on precision is only discussed qualitatively. A brief quantitative justification or sensitivity check would strengthen the methodological rigor.

Justification for SMOTE should be clearer.

While SMOTE is adopted as the final imbalance-handling strategy, comparative results against alternative approaches such as class-weighted learning are not shown. A concise comparison would improve transparency.

ROC–AUC interpretation should be more cautious.

The discussion of the achieved AUC would benefit from clearer framing as being consistent with prior crash severity studies, rather than implying a general performance benchmark.

SHAP interpretations should emphasize non-causality.

Although the SHAP analysis is informative, some interpretations may be influenced by reporting or behavioral factors. A clearer distinction between associative patterns and actionable mechanisms would improve clarity.

6. PLOS authors have the option to publish the peer review history of their article (what does this mean?). If published, this will include your full peer review and any attached files.). If published, this will include your full peer review and any attached files.). If published, this will include your full peer review and any attached files.). If published, this will include your full peer review and any attached files.

...

Reviewer #1: No

Reviewer #2: No

---

## [Author Response · Author response to Decision Letter 1]

23 Mar 2026

Detailed response to Editor and Reviewer’s Comments

Manuscript No.: PONE-D-26-00177

Title: Safety-Oriented and Explainable Machine Learning for KSI Crash Risk Prediction: Evidence from the United Kingdom.

Authors: Khanh Giang Le

First, I would like to thank the editor for considering my research and the reviewers for your highly constructive comments, which have helped me to enhance the manuscript. I sincerely appreciated all the suggestions made by the editor and the reviewers. I have carefully reviewed these comments and have substantially revised and expanded the manuscript in accordance with the suggestions in order to improve its content and overall quality. The changes made to the manuscript are highlighted using different colors.

Academic Editor Comments=================

Comment: After careful consideration, we feel that it has merit but does not fully meet PLOS ONE’s publication criteria as it currently stands. Therefore, we invite you to submit a revised version of the manuscript that addresses the points raised during the review process.

Answer: I sincerely thank you and the reviewers for your time and careful evaluation of my manuscript. I greatly appreciate the constructive and insightful comments, which have helped improve the quality of the paper. I have carefully considered all comments and suggestions from both the editor and reviewers, and have thoroughly revised the manuscript accordingly. All changes have been made to address the concerns raised and to ensure that the manuscript meets the journal’s publication standards.

Comment 1: Please ensure that your manuscript meets PLOS ONE's style requirements, including those for file naming. The PLOS ONE style templates can be found at

Answer: Thank you for this comment. The manuscript has been revised to comply with the PLOS ONE formatting and style requirements, including file naming conventions, as specified in the journal guidelines and templates.

Comment 2: Please note that PLOS One has specific guidelines on code sharing for submissions in which author-generated code underpins the findings in the manuscript. In these cases, we expect all author-generated code to be made available without restrictions upon publication of the work. Please review our guidelines at https://journals.plos.org/plosone/s/materials-and-software-sharing#loc-sharing-code and ensure that your code is shared in a way that follows best practice and facilitates reproducibility and reuse.

Answer: Thank you for this comment. In accordance with the PLOS ONE guidelines on code sharing, all author-generated code underlying the findings of this study has been made publicly available without restriction. The code has been deposited in a public repository (Zenodo) and can be accessed at: [https://doi.org/10.5281/zenodo.19135623]. This ensures full transparency, reproducibility, and reuse of the results.

Comment 3: Thank you for uploading your study's underlying data set. Unfortunately, the repository you have noted in your Data Availability statement does not qualify as an acceptable data repository according to PLOS's standards. At this time, please upload the minimal data set necessary to replicate your study's findings to a stable, public repository (such as figshare or Dryad) and provide us with the relevant URLs, DOIs, or accession numbers that may be used to access these data. For a list of recommended repositories and additional information on PLOS standards for data deposition, please see https://journals.plos.org/plosone/s/recommended-repositories.

Answer: Thank you for this comment. In response to the journal’s data sharing requirements, the underlying dataset has been uploaded to a stable and publicly accessible repository that complies with PLOS ONE standards. The dataset is available at: [https://doi.org/10.5281/zenodo.19135778]. The deposited data include the minimal dataset required to replicate the findings reported in this study. The Data Availability Statement has been updated accordingly to ensure full transparency and reproducibility.

Comment 4: We note you have included a table to which you do not refer in the text of your manuscript. Please ensure that you refer to Table 3 in your text; if accepted, production will need this reference to link the reader to the Table.

Answer: Thank you for this comment. The manuscript has been revised to include an explicit reference to Table 3 in the main text. This ensures proper linkage between the text and the table for clarity and consistency.

Comment 5: If the reviewer comments include a recommendation to cite specific previously published works, please review and evaluate these publications to determine whether they are relevant and should be cited. There is no requirement to cite these works unless the editor has indicated otherwise.

Answer: Thank you for this comment. I have carefully reviewed the recommended references and found them to be relevant to the scope of this study. Accordingly, these references have been incorporated into the revised manuscript where appropriate to strengthen the literature context.

Comment 6: Please review your reference list to ensure that it is complete and correct. If you have cited papers that have been retracted, please include the rationale for doing so in the manuscript text, or remove these references and replace them with relevant current references. Any changes to the reference list should be mentioned in the rebuttal letter that accompanies your revised manuscript. If you need to cite a retracted article, indicate the article’s retracted status in the References list and also include a citation and full reference for the retraction notice.

Answer: Thank you for this comment. The reference list has been carefully reviewed and updated to ensure completeness and accuracy. No retracted articles are included in the revised manuscript. Any necessary corrections and updates to the references have been made accordingly.

Reviewer #1=======================

Comment: Overall Recommendation: Minor Revision

The manuscript proposes a safety-oriented and explainable machine learning (ML) framework for predicting Killed or Seriously Injured (KSI) crashes using nationwide UK traffic accident data from 2020 to 2024. By framing the problem as a binary classification (KSI vs. slight injury) and systematically addressing class imbalance, decision threshold optimisation, probability calibration, and model interpretability, the study offers a meaningful contribution to the field of traffic safety analytics. However, several issues should be addressed to strengthen before final acceptance.

Answer: I sincerely thank the reviewer for the positive evaluation of my study and for recognising the potential contribution of the proposed safety-oriented and explainable machine learning framework to traffic safety analysis. I appreciate the reviewer’s constructive comments and suggestions. All comments have been carefully considered, and the manuscript has been revised accordingly to address the issues raised. All additions and revisions in the revised manuscript have been highlighted in green for clarity. Detailed responses to each comment are provided below.

Comment 1: Please clarify whether feature selection was performed before modelling.

Answer: Thank you very much for your constructive and insightful comments.

I have clarified the feature selection procedure in Section (Data Preprocessing and Feature Engineering) of the revised manuscript. Feature selection was conducted prior to model training using a domain-informed filtering strategy. Specifically, non-informative administrative identifiers, redundant spatial coordinate systems, and post-crash outcome variables were removed to prevent information leakage and ensure that only explanatory variables available before crash occurrence were used in model training. The revised manuscript now explicitly describes these filtering criteria and provides representative examples of the excluded variables.

Changes made in the revised manuscript (Section: Data Preprocessing and Feature Engineering):

…“Feature selection was conducted prior to model training using a domain-informed filtering strategy. This filtering process was designed to eliminate variables that may introduce information leakage and to retain interpretable predictors relevant to traffic safety analysis.

Specifically, variables were systematically evaluated and removed based on three criteria. First, administrative identifiers (e.g., collision_index and collision_ref_no) were excluded because they are non-informative record labels that do not contain meaningful predictive information for crash severity modeling. Second, redundant coordinate systems (e.g., location_easting_osgr and location_northing_osgr) were removed to avoid duplicate spatial representations. Standard geographic coordinates (latitude and longitude) were retained as the primary spatial references, allowing the tree-based model to learn broader spatial patterns and regional safety variations. Finally, post-crash outcome variables (e.g., number_of_casualties and collision_adjusted_severity_serious) were excluded. These post-event variables represent consequences of the crash rather than predictive risk factors, and their inclusion would introduce data leakage into the modeling process.

After this domain-driven filtering, all remaining variables—representing roadway characteristics, environmental conditions, and temporal attributes—were retained without further feature elimination. To further enhance model learning, temporal variables were decomposed into interpretable components (month and hour of occurrence) to capture seasonal and diurnal traffic patterns. Missing values were subsequently addressed using mode imputation for categorical attributes and median imputation for numerical ones [22]. This preprocessing ensures data consistency and improves model robustness.”…

Comment 2: Please clarify the hyperparameter settings for the adopted models.

Answer: Thank you very much for your constructive and insightful comments. I agree that clearly specifying the hyperparameter settings is important for transparency and reproducibility.

In the revised manuscript, I have added a detailed description of the model configurations in Section (Model Development Using LightGBM). The manuscript has been revised accordingly.

Specifically, for the proposed LightGBM model, the main parameters include a learning rate of 0.03, 1000 boosting iterations, num_leaves = 40, an unrestricted maximum depth (max_depth = −1), and early stopping based on validation loss with 100 rounds.

For comparative evaluation, the baseline models were configured as follows: Logistic Regression with L2 regularization and max_iter = 1000; Random Forest with 100 trees and max_depth = 20; and XGBoost with 1000 boosting rounds, learning_rate = 0.03, max_depth = 6, and tree_method = "hist". These settings ensure a consistent and fair comparison across models.

Changes made in the revised manuscript (Section: Model Development Using LightGBM):

…“To rigorously evaluate the effectiveness of the proposed LightGBM model, several widely used machine learning algorithms, namely Logistic Regression, Random Forest, and XGBoost, were implemented as baseline models for comparative analysis.

To ensure reproducibility and a fair comparison across all algorithms, the hyperparameters were explicitly configured. For the proposed LightGBM model, the main parameters included a learning rate of 0.03, 1000 boosting iterations, and a tree structure controlled by num_leaves = 40 with an unrestricted maximum depth (max_depth = −1).

For the baseline models, configurations were set as follows: Logistic Regression was implemented with L2 regularization and max_iter = 1000; Random Forest used 100 trees with a maximum depth of 20; and XGBoost was trained with 1000 boosting rounds, learning_rate = 0.03, max_depth = 6, and tree_method = "hist".

These configurations follow commonly adopted settings in tabular machine learning tasks, ensuring both model stability and fair comparison across algorithms.”…

Comment 3: The reported ROC-AUC value is 0.664. The authors could provide a more apparent justification for why this level is sufficient for safety-critical applications, perhaps by citing similar studies in the literature.

Answer: Thank you very much for your constructive and insightful comments. I fully agree that providing contextual justification for the reported ROC–AUC value is important, particularly for safety-critical applications.

In the revised manuscript, I have added a new subsection entitled “Contextual Interpretation of Model Performance” to clarify this point and to position the model performance within the broader crash severity modeling literature. Specifically, I explain that crash severity prediction is inherently challenging due to the stochastic and highly heterogeneous nature of traffic accidents. As discussed by Savolainen et al. (2011), crash outcomes are influenced by numerous interacting factors and substantial unobserved heterogeneity, which naturally limits the achievable predictive accuracy of statistical and machine learning models.

To further contextualize the obtained performance, I also cite recent empirical studies applying machine learning to crash severity prediction. For example, Komol et al. (2021) reported test accuracies ranging from 64.45% to 72.30% when applying Random Forest, SVM, and KNN models to vulnerable road user crash severity classification. Similarly, a recent LightGBM-based fatality prediction study (Rifat et al., 2024) reported a maximum ROC–AUC of approximately 0.72. These findings indicate that predictive performance in the range of AUC ≈ 0.60–0.72 is commonly observed for real-world crash severity modeling problems.

Furthermore, my study adopts a safety-oriented modeling strategy, in which the decision threshold is optimized (τ = 0.35) to improve the detection of severe crashes. As a result, the proposed LightGBM model achieves a Recall of 0.605 for the KSI (Killed or Seriously Injured) class, meaning that more than 60% of severe crashes are correctly identified. In practical traffic safety management, minimizing false negatives (i.e., failing to detect high-risk crashes) is often more critical than marginal improvements in global discrimination metrics.

These explanations and supporting references have been added in (Section: Contextual Interpretation of Model Performance) of the revised manuscript to clarify why the obtained ROC–AUC value is consistent with the expected performance range for crash severity prediction tasks.

Changes made in the revised manuscript (Section: Contextual Interpretation of Model Performance):

“Contextual Interpretation of Model Performance

Crash severity prediction is inherently challenging due to the complex and stochastic nature of road traffic crashes. Unlike many prediction tasks in controlled domains, traffic accidents are influenced by numerous interacting factors, including roadway characteristics, driver behavior, vehicle conditions, environmental factors, and random circumstances occurring at the moment of collision. As highlighted by Savolainen et al. (2011), crash severity outcomes are strongly affected by unobserved heterogeneity and behavioral uncertainty, which significantly limits the achievable predictive accuracy of statistical and machine learning models [26]. Consequently, moderate predictive performance is commonly reported in the crash severity literature.

Empirical evidence from recent machine learning studies supports this observation. For instance, Komol et al. (2021) applied Random Forest, Support Vector Machine, and K-Nearest Neighbor models to analyze crash severity of vulnerable road users using Queensland crash data and reported test accuracies ranging from 64.45% to 72.30%, depending on the road user group considered [31]. Similarly, Rifat et al. (2024) reported that a recent LightGBM-based fatality prediction model achieved a maximum ROC–AUC

---

## [Decision Letter · Decision Letter 1]

8 Apr 2026

Safety-Oriented and Explainable Machine Learning for KSI Crash Risk Prediction: Evidence from the United Kingdom

PONE-D-26-00177R1

Dear Dr. Le,

We’re pleased to inform you that your manuscript has been judged scientifically suitable for publication and will be formally accepted for publication once it meets all outstanding technical requirements.

Kind regards,

Xuecai Xu, Ph.D.

Academic Editor

PLOS One

Additional Editor Comments (optional):

Please polish the language before finally being published online.

Reviewers' comments:

Reviewer's Responses to Questions

**Comments to the Author**

1. If the authors have adequately addressed your comments raised in a previous round of review and you feel that this manuscript is now acceptable for publication, you may indicate that here to bypass the “Comments to the Author” section, enter your conflict of interest statement in the “Confidential to Editor” section, and submit your "Accept" recommendation.

Reviewer #1: All comments have been addressed

2. Is the manuscript technically sound, and do the data support the conclusions?

Reviewer #1: Yes

3. Has the statistical analysis been performed appropriately and rigorously? 

Reviewer #1: Yes

4. Have the authors made all data underlying the findings in their manuscript fully available?

The PLOS Data policy requires authors to make all data underlying the findings described in their manuscript fully available without restriction, with rare exception (please refer to the Data Availability Statement in the manuscript PDF file). The data should be provided as part of the manuscript or its supporting information, or deposited to a public repository. For example, in addition to summary statistics, the data points behind means, medians and variance measures should be available. If there are restrictions on publicly sharing data—e.g. participant privacy or use of data from a third party—those must be specified.requires authors to make all data underlying the findings described in their manuscript fully available without restriction, with rare exception (please refer to the Data Availability Statement in the manuscript PDF file). The data should be provided as part of the manuscript or its supporting information, or deposited to a public repository. For example, in addition to summary statistics, the data points behind means, medians and variance measures should be available. If there are restrictions on publicly sharing data—e.g. participant privacy or use of data from a third party—those must be specified.requires authors to make all data underlying the findings described in their manuscript fully available without restriction, with rare exception (please refer to the Data Availability Statement in the manuscript PDF file). The data should be provided as part of the manuscript or its supporting information, or deposited to a public repository. For example, in addition to summary statistics, the data points behind means, medians and variance measures should be available. If there are restrictions on publicly sharing data—e.g. participant privacy or use of data from a third party—those must be specified.requires authors to make all data underlying the findings described in their manuscript fully available without restriction, with rare exception (please refer to the Data Availability Statement in the manuscript PDF file). The data should be provided as part of the manuscript or its supporting information, or deposited to a public repository. For example, in addition to summary statistics, the data points behind means, medians and variance measures should be available. If there are restrictions on publicly sharing data—e.g. participant privacy or use of data from a third party—those must be specified.

Reviewer #1: Yes

5. Is the manuscript presented in an intelligible fashion and written in standard English?

Reviewer #1: Yes

6. Review Comments to the Author

Reviewer #1: (No Response)

7. PLOS authors have the option to publish the peer review history of their article (what does this mean?). If published, this will include your full peer review and any attached files.). If published, this will include your full peer review and any attached files.). If published, this will include your full peer review and any attached files.). If published, this will include your full peer review and any attached files.

...

Reviewer #1: No

---

## [Editor Report · Acceptance letter]

PONE-D-26-00177R1

PLOS One

Dear Dr. Le,

I'm pleased to inform you that your manuscript has been deemed suitable for publication in PLOS One. Congratulations! Your manuscript is now being handed over to our production team.

Kind regards,

on behalf of

Dr. Xuecai Xu

Academic Editor

PLOS One